# Association of *IL-4* and *IL-4R* Polymorphisms with Litter Size Traits in Pigs

**DOI:** 10.3390/ani11041154

**Published:** 2021-04-17

**Authors:** Worrarak Norseeda, Guisheng Liu, Tawatchai Teltathum, Pantaporn Supakankul, Korawan Sringarm, Watcharapong Naraballobh, Trisadee Khamlor, Siriwadee Chomdej, Korakot Nganvongpanit, Patcharin Krutmuang, Supamit Mekchay

**Affiliations:** 1Department of Animal and Aquatic Sciences, Faculty of Agriculture, Chiang Mai University, Chiang Mai 50200, Thailand; worrarak_n@cmu.ac.th (W.N.); korawan.s@cmu.ac.th (K.S.); watcharapong.n@cmu.ac.th (W.N.); trisadee.kha@cmu.ac.th (T.K.); 2Center of Excellence on Agricultural Biotechnology: (AG-BIO/PERDO-CHE), Bangkok 10900, Thailand; 3Graduate School, Chiang Mai University, Chiang Mai 50200, Thailand; 4Institute of Animal Science and Veterinary Medicine, Hubei Academy of Agricultural Sciences, Wuhan 430064, China; guisheng_liu1964@yahoo.com; 5Hubei Key Lab for Animal Embryo Engineering and Molecular Breeding, Wuhan 430064, China; 6Mae Hong Son Livestock Research and Breeding Center, Mae Hong Son 58000, Thailand; Ismh_mhs@dld.go.th; 7Division of Animal Science, School of Agriculture and Natural Resources, University of Phayao, Phayao 56000, Thailand; pantaporn.su@up.ac.th; 8Cluster of Research and Development of Pharmaceutical and Natural Products Innovation for Human or Animal, Chiang Mai University, Chiang Mai 50200, Thailand; 9Innovative Agriculture Research Center, Faculty of Agriculture, Chiang Mai University, Chiang Mai 50200, Thailand; patcharin.k@cmu.ac.th; 10Department of Biology, Faculty of Science, Chiang Mai University, Chiang Mai 50200, Thailand; siriwadee.ch@cmu.ac.th; 11Department of Veterinary Bioscience and Veterinary Public Health, Faculty of Veterinary Medicine, Chiang Mai University, Chiang Mai 50100, Thailand; korakot.n@cmu.ac.th; 12Department of Entomology and Plant Pathology, Faculty of Agriculture, Chiang Mai University, Chiang Mai 50200, Thailand

**Keywords:** *IL-4*, *IL-4R*, litter size, pig, SNPs

## Abstract

**Simple Summary:**

The *IL-4* and *IL-4R* cytokine genes are responsible for immune response in the reproductive system and are related to embryonic implantation and fetal survival during pregnancy in females. However, to date, their effects on litter size traits in pigs have been not elucidated. Therefore, the present study was conducted to verify the porcine *IL-4* and *IL-4R* polymorphisms and assess how they affect litter size traits in commercial pigs. The findings suggested that the porcine *IL-4* g.134993898T > C and *IL-4R* c.1577A > T polymorphisms were associated with the litter size traits. Therefore, the porcine *IL-4* and *IL-4R* genes may be potentially effective genetic markers to improve the litter size traits in pigs.

**Abstract:**

The interleukin-4 (IL-4) and interleukin-4 receptor (IL-4R) are cytokines that are involved in the immune and reproductive systems. This study aimed to verify the polymorphisms in the porcine *IL-4* and *IL-4R* genes and to assess their effects on litter size traits in commercial pigs. Single nucleotide polymorphisms (SNPs) in the porcine *IL-4* and *IL-4R* genes were genotyped by the polymerase chain reaction-restriction fragment length polymorphism (PCR-RFLP) method. A non-coding SNP of *IL-4* g.134993898T > C and a non-synonymous SNP of *IL-4R* c.1577A > T (amino acid change at position 526, Q526L) were found to be segregating in Landrace sows. The *IL-4* g.134993898T > C polymorphism was significantly associated with the number of piglets weaned alive (NWA) trait. The *IL-4R* c.1577A > T polymorphism was significantly associated with the number born alive (NBA) and NWA traits. Moreover, the accumulation of favorable alleles of these two SNP markers revealed significant associations with the NBA, NWA, and mean weight of piglets at weaning (MWW) traits. These findings indicate that the porcine *IL-4* and *IL-4R* genes may contribute to the reproductive traits of pigs and could be used as candidate genes to improve litter size traits in the pig breeding industry.

## 1. Introduction

Litter size traits are among the most important traits for breeding in commercial pig production. Increasing litter size is of great economic interest as a means of enhancing the productivity of the pig industry [1,2,3]. Major limitations to increasing litter size in pigs are embryonic mortality and fetal losses during the pregnancy period [4,5]. Successful pregnancy depends on coordinated and precisely regulated cellular and molecular processes of the conceptus and the maternal endometrium for the establishment of pregnancy. Communication between the conceptus and the endometrium is mediated by cytokines and cell surface receptors [6], and it is essential for embryo implantation, placental development, and the effective maintenance of pregnancy [7,8]. Several cytokine genes involved in embryo attachment and implantation that affect the relevant litter size traits have also been identified in pigs, e.g., erythropoietin-producing hepatocellular A4 (*EphA4*) [5], ring finger protein 4 gene (*RNF4*) [9], leukemia inhibitory factory (*LIF*) [10], interleukin-6 (*IL-6*) [11], and osteopontin (*OPN*) [12]. Currently, numerous studies demonstrate that interleukin-4 (IL-4) and its receptor (IL-4 receptor, IL-4R) play an important role in embryo implantation and pregnancy in pigs [13,14] and humans [15].

The IL-4 is an anti-inflammatory cytokine that is mainly produced by T-helper (Th) 2 cells, natural killer (NK) cells, mast cells, and basophils [16]. It is involved in the regulation of immune system responses [17]. The expression levels of *IL-4* mRNAs are increased in the uterine endometrium during implantation in pigs and humans [13,14,15]. It has been documented that a deficiency of IL-4 cytokine can lead to infertility and various pregnancy disorders in mammals [18]. The *IL-4* gene has been mapped on the long arm of the *Sus scrofa* chromosome 2 (SSC2q) and is composed of four exons and three introns that are encoded with a peptide of 133 amino acids (ENSSSCG00050075333; Ensembl Sscrofa 11.1; https://asia.ensembl.org/Sus_scrofa/Info/Index, accessed on 2 September 2020). Besides, the porcine *IL-4* gene is closely located (SSC2, 134.9 Mb) to the quantitative trait loci (QTL) regions for ovulation rates (128.8–145.1 Mb), total number born (139.9–140.9 Mb), mummified fetuses (133.4–145.1 Mb), numbers of stillborn (128.5–137.1 Mb), the mortality of piglets (138.7–143.7 Mb), gestation length (153.0 Mb), and age at puberty (133.3–157.3 Mb) [19,20,21,22]. Polymorphisms of the porcine *IL-4* gene have been characterized and reported in the Ensembl database (https://asia.ensembl.org/Sus_scrofa/Info/Index, accessed on 2 September 2020). Moreover, associations of the *IL-4* polymorphisms with the recurrent abortion, pregnancy specific-disorder of pre-eclampsia, allergic diseases, and serum immunoglobulin E (IgE) levels have been reported in humans and mice [23,24,25,26,27]. These pieces of evidence suggest that the *IL-4* gene is involved in implantation and pregnancy, thus the *IL-4* gene is important to litter size traits.

The *IL-4R* is an immune gene that encodes the alpha-chain of the IL-4R molecule, a type-I transmembrane protein that can bind IL-4 and IL-13, and mediates its effect through kinases of the Janus kinase (JAK) family, leading to tyrosine phosphorylation of several substrates in cells [28,29]. The *IL-4R* mRNAs are expressed in the endometrium and conceptus [30,31]. The *IL-4R* gene has been mapped on *Sus scrofa* chromosome 3 (SSC3). It contains ten exons and nine introns that are encoded with a peptide of 824 amino acids (ENSSSCG00000007817; Ensembl Sscrofa 11.1; https://asia.ensembl.org/Sus_scrofa/Info/Index, accessed on 2 September 2020). Moreover, the porcine *IL-4R* gene is closely located (SSC3, 19.5 Mb) to the QTL regions for total number born (15.3–20.0 Mb), litter birth intervals (11.9–33.7 Mb), and age at puberty (11.1–27.20 Mb) [20,22,32]. The polymorphisms of the porcine *IL-4R* gene have been characterized and reported in the Ensembl database (https://asia.ensembl.org/Sus_scrofa/Info/Index, accessed on 2 September 2020). Moreover, the polymorphisms of *IL4R* gene are associated with the pregnancy disorders and various cancer types in humans [27,33,34,35,36].

All these shreds of evidence suggest that the *IL-4* and *IL-4R* genes are responsible for immune response in the reproductive system of mammals. Moreover, their functions are critical for embryonic implantation and fetal survival during pregnancy, as well as their positions, which are closely located to the QTLs for reproductive traits in pigs. Therefore, the *IL-4* and its receptor (*IL-4R*) genes can be regarded as positional and functional candidate genes for the determination of the reproductive traits of pigs. However, information on the association of the *IL-4* and *IL-4R* genes with litter size traits in pigs has been limited. Thus, the porcine *IL-4* and *IL-4R* SNPs were selected based on their segregation in the pig population to elucidate their association with litter size traits. In the present study, we have verified the polymorphisms in porcine *IL-4* and *IL-4R* genes, while their association with litter size traits was assessed in commercial pigs.

## 2. Materials and Methods

### 2.1. Animals and DNA Extraction

Blood samples were taken from a total of 323 sows of the Landrace pig breed. These Landrace breeding stocks were obtained from a commercial nucleus herd that is established for improved growth performance and reproductive traits. All sows were reared under commercial conditions of the Betagro Hybrid International Company, Thailand. These sows were kept in closed houses with an evaporative cooling system and were fed a corn–soybean-based diet containing 16% crude protein and 3388 kcal/kg digestible energy. The reproductive performance traits of the sows were assessed in 1162 litters (1 to 8 parities) and were recorded in terms of litter size. These traits consisted of total number born (TNB), number born alive (NBA), the number of piglets weaned alive (NWA), mean birth weight of the piglets (MBW), and mean weight of piglets at weaning (21 days, MWW). Genomic DNA was extracted from blood samples using the Chelex method [37] and kept at 4 °C until analysis.

### 2.2. Verification of Porcine IL-4 and IL-4R Polymorphisms and Genotyping

To verify the SNPs in the porcine *IL-4* gene, specific primers were designed based on the available nucleotide sequence information (GenBank accession number: NC_010444.3) to cover four exons and three introns of the porcine *IL-4* gene, as shown in Table 1. Ten DNA samples were selected from the Landrace population with the five highest and five lowest TNB values and were used to amplify the DNA segments of the porcine *IL-4* gene using each primer (Table 1). The amplicons of the porcine *IL-4* gene were sequenced using the CEQ 8000 Genetic Analysis System (Beckman-Coulter, Brea, CA, USA) to find out the SNPs segregating in this pig population. To verify the SNPs in the porcine *IL-4R* gene, five non-synonymous SNPs (c.163G > A, c.242C > T, c.623G > A, c.1016G > T, and c.1577A > T) of the porcine *IL-4R* gene were selected based on the restriction enzymes available in the Ensembl database (ENSSSCT00000008566.4; http://asia.ensembl.org/index.html, accessed on 2 September 2020). These were used to verify the SNP in the Landrace pig population. The specific primers of the porcine *IL-4R* gene were designed based on relevant nucleotide sequence information (GenBank accession number: NC_010445.3), as shown in Table 2. The verified SNPs of the porcine *IL-4* and *IL-4R* genes were used purposely for association analysis in this study. These SNPs were genotyped using the polymerase chain reaction-restriction fragment length polymorphism (PCR-RFLP) assay. The PCR amplification was performed in a total volume of 20 μL consisting of 50 ng of a genomic DNA sample, 1× (NH_4_)_2_SO_4_ buffer, 1.5 mM MgCl_2_, 0.2 mM dNTPs, 0.4 μM each primer (Table 2), and 0.2 U *Taq* DNA polymerase (Fermentas, Hanover, MD, USA). The PCR conditions were as follows: 94 °C for 3 min at the initial denaturing stage; followed by 35 cycles of 94 °C for 30 s, 58–60 °C for 30 s, and 72 °C for 30 s; and then 5 min at 72 °C to complete the reaction. The PCR products were digested with 2.5 U of the restriction enzyme (Fermentas, Hanover, MD, USA) for each fragment (Table 2) and incubated for 2 h. The digested PCR products were then separated on 6% polyacrylamide electrophoresis in 1× TBE (Tris-borate-EDTA) buffer and visualized by ethidium bromide staining.

### 2.3. Statistical Analysis

Allelic and genotypic frequencies were calculated. Hardy–Weinberg equilibrium (HWE) was analyzed using the chi-square test. Association analysis of porcine *IL-4* and *IL-4R* SNP markers with the litter size traits was examined using a general linear model. The statistical model was used as follows: Y*_ijkl_* = μ + P*_i_* + YS*_j_* + G*_k_* + e*_ijkl_*, where Y*_ijkl_* is representative of the observations of the phenotype values, μ represents the average normalized record of populations, P*_i_* represents the fixed effect of parities (*i* = 1 and ≥2), YS*_j_* represents the fixed effect of year-season (*j* = 1–8), G*_k_* is representative of the fixed effect of the genotypes for *IL-4* and *IL-4R* (*k* = 1–3) or the accumulated favorable alleles for *IL-4* and *IL-4R* (*k* = 0–4), and e*_ijkl_* represents the residual error. Moreover, the additive effect was calculated as the half difference between the two homozygous genotypes and the dominance effect was estimated as the deviation of the heterozygous genotype effect from the mean effect of the two homozygous genotypes [38]. The estimated effects were calculated using a *t*-test on significant deviations from zero. Furthermore, linear regression between the number of favorable alleles of porcine *IL-4* and *IL-4R* genes and least square mean values of the litter size traits were analyzed.

## 3. Results

### 3.1. Polymorphisms of Porcine IL-4 and IL-4R Genes

To verify the polymorphisms of the porcine *IL-4* gene, the DNA segments of exons and introns of the porcine *IL-4* gene were sequenced. An SNP in intron 3 of the porcine *IL-4* gene was found and corresponded to an SNP of porcine *IL-4* g.134993898T > C (rs329453960) locus in Ensembl data (https://asia.ensembl.org/Sus_scrofa/Info/Index, accessed on 2 September 2020). It was detected with the restriction enzyme *Bsu*RI. Two specific alleles revealed two fragments of 295 and 38 bp for allele T and three fragments of 180, 115, and 38 bp for allele C (Figure 1A). To verify the polymorphisms of the porcine *IL-4R* gene, five non-synonymous SNPs were selected to be examined in the Landrace pig population. A polymorphic site of c.1577A > T (rs342791614) was found in exon 8. It was a non-synonymous mutation leading to a non-conservative amino acid exchange at position 526 from glutamine to leucine (Q526L). This polymorphic site was detected with the restriction enzyme *Alu*I. Two specific alleles revealed a 137 bp fragment for allele A and two fragments of 119 and 18 bp for allele T (Figure 1B). However, no polymorphisms of the four SNP markers (c.163G > A, c.242C > T, c.623G > A, and c.1016G > T) of the porcine *IL-4R* gene were observed in this study.

### 3.2. Genotypic and Allelic Frequencies

The genotypic and allelic frequencies of the porcine *IL-4* and *IL-4R* genes are shown in Table 3. Two polymorphic sites of the porcine *IL-4* g.134993898T > C and *IL-4R* c.1577A > T genes were found to be segregating in the Landrace sows. At the *IL-4* g.134993898T > C and *IL-4R* c.1577A > T loci, three genotypes were observed. The *IL-4* g.134993898T and *IL-4R* c.1577A alleles were more frequent in this pig population. Moreover, the four SNP markers of the porcine *IL-4R* gene at c.163, c.242, c.623, and c.1016 loci were fixed as c.163G, c.242C, c.623G, and c.1016G, respectively (data not shown). The chi-square test revealed that the genotype distributions of the porcine *IL-4* g.134993898T > C and *IL-4R* c.1577A > T loci within Landrace sows were in agreement with the HWE specifications (*p* > 0.05).

### 3.3. Associations of Porcine IL-4 and IL-4R Polymorphisms with Litter Size Traits

The effects of porcine *IL-4* and *IL-4R* genotypes on litter size traits were assessed in the Landrace sows. The porcine *IL-4* and *IL-4R* polymorphisms were associated with litter size traits of pigs. Association of the porcine *IL-4* g.134993898T > C polymorphism with litter size traits is shown in Table 4. No significant association of porcine *IL-4* g.134993898T > C polymorphism with any litter size traits was found in the first parity of sows. However, the porcine *IL-4* g.134993898T > C polymorphism was significantly associated with the NWA trait in the later parities of sows. The sows with the TT and TC genotypes had higher NWA values than the sows with the CC genotype. The significant additive effect for the NWA trait was detected in the later parities of sows. Thus, the porcine *IL-4* g.134993898T allele seems to be a favorable allele for litter size traits in pigs. Association of the porcine *IL-4R* c.1577A > T polymorphism with litter size traits is shown in Table 5. There was no significant association of porcine *IL-4R* c.1577A > T polymorphism with any litter size traits in the first parity of sows. However, the porcine *IL-4R* c.1577A > T polymorphism was significantly associated with the NBA and NWA traits in later parities of sows. Notably, the sows with the TT genotype had higher NBA and NWA values than the sows with the AA and TA genotypes. Additionally, the significant additive effects for the NBA and NWA traits were observed in later parities of sows. Thus, the porcine *IL-4R* c.1577T allele seems to be a favorable allele for litter size traits in pigs.

The effects of the accumulated favorable alleles of porcine *IL-4* g.134993898T > C and *IL-4R* c.1577A > T on litter size traits are shown in Table 6. No significant association of the accumulated favorable alleles with litter size traits was observed in the first parity of sows. In later parities, significant associations of the accumulated favorable alleles with the NBA, NWA, and MWW traits were found. Furthermore, the results of linear regression analysis revealed that the increased accumulated favorable alleles were highly positively correlated with the least square mean values of the TNB, NBA, and NWA traits in later parities of sows (Figure 2A–C). However, lower correlations between the accumulated favorable alleles and the least square mean values of litter size traits were observed in the first parity, as well as the MBW and MWW traits in the later parities of sows (result not shown). Notably, the largest number of favorable alleles (TTTT) of porcine *IL-4* and *IL-4R* genotypes seemed to have advantageous effects on litter size traits. On the other hand, the smallest number of favorable alleles (CCAA) of porcine *IL-4* and *IL-4R* genotypes seemed to have disadvantageous effects on these traits.

## 4. Discussion

Increasing litter size is of significant economic interest in the pig industry [1,2,3]. Embryonic survival is the main factor in determining litter size in pigs and depends on maternal recognition of pregnancy, embryo attachment, implantation, placental development, uterine capacity, and the maintenance of pregnancy [4,39]. Numerous studies have demonstrated the role of different cytokines in the channel of communication established between the trophoblast and maternal endometrium, which contributes to embryo implantation and the pregnancy of pigs [8,40,41].

IL-4 is an anti-inflammatory Th2 cytokine that can bind IL-4R to regulate IgE production, immune response, and embryonic implantation [13,29,42]. Several studies have reported that polymorphisms of *IL-4* and/or *IL-4R* genes are associated with allergic diseases [23,24], diabetes [43], cancers [36], and pregnancy disorders [25,26,27]. In the present study, we have verified and elucidated polymorphisms of the porcine *IL-4* and *IL-4R* genes with respect to reproductive performance traits in commercial pigs. The porcine *IL-4* g.134993898T > C and *IL-4R* c.1577A > T polymorphisms were found to be segregating in Landrace sows. At the porcine *IL-4* g.134993898T > C and *IL-4R* c.1577A > T loci, three possible genotypes were observed in pigs. The *IL-4* g.134993898T and *IL-4R* c.1577A were major alleles in this pig population. The results of the chi-square test revealed that the porcine *IL-4* g.134993898T > C and *IL-4R* c.1577A > T polymorphisms in these sows met the HWE specifications. This result indicates that the porcine *IL-4* g.134993898T > C and *IL-4R* c.1577A > T polymorphisms were under homeostasis when accompanied by the effects of artificial selection.

The porcine *IL-4* is mapped on SSC2 at position 134.9 Mb. Recently, a conducted genome-wide association study (GWAS) revealed that the porcine *IL-4* gene is located within QTLs regions for the TNB (139.9–140.9 Mb) and piglet mortality traits (138.7–143.7 Mb) in pigs [21]. Moreover, the expression levels of the porcine *IL-4* gene are upregulated in the endometrium during implantation [14] and involved in the gene networks for pregnancy establishment in pigs [13]. Furthermore, the porcine IL-4 protein concentrations are increased in serum of pregnant sows, as well as in maternal and fetal placenta during pregnancy periods [8]. These evidences indicate that the *IL-4* gene plays a crucial role in embryo implantation and may be important for litter size traits.

In this study, the polymorphism of the porcine *IL-4* gene had a significant association with litter size traits in pigs. A positive effect of the favorable *IL-4* g.134993898T allele on litter size traits was found. The *IL-4* g.134993898T > C polymorphism was found to be located in the non-coding sequence of the porcine *IL-4* gene. We hypothesize that this SNP might be in linkage disequilibrium with other causal polymorphisms that may be located in another region of the porcine *IL-4* gene. The association of the porcine *IL-4* gene with litter size traits may be attributed to the balance of cytokines in endometrium during implantation period and the regulation of endometrial cytokines by IL-4. The *IL-4* polymorphisms may be implicated in the impaired Th1/Th2 cytokine balance and may influence implantation defects and pregnancy disorders. Previous studies have demonstrated that increased anti-inflammatory Th2 cytokines (e.g., IL-4, IL-10) and decreased pro-inflammatory Th1 cytokines (e.g., IL-2, TNF-α, IFN-γ) levels in endometrium are important for successful pregnancy [6,15]. However, the increased Th1/Th2 cytokine ratios may increase cytotoxicity against embryos and lead to implantation failure [44,45]. Moreover, the *IL-4* polymorphisms are associated with the recurrent abortion and pregnancy specific disorder of pre-eclampsia in humans [26,27]. Furthermore, the *IL-4* polymorphisms may influence the other endometrial cytokines in promoting embryo implantation and placental development during pregnancy. Previous studies have reported that the endometrial cytokines of leukemia inhibitory factor (LIF) and vascular endothelial growth factor (VEGF) are upregulated by IL-4 [46,47,48]. The LIF is an endometrial requirement for implantation and embryo development [47] and the VEGF is involved in endometrial angiogenesis [14]. The decreased *VEGF* and *IL-4* expression levels are associated with recurrent spontaneous miscarriage or recurrent abortion in women [49], and the decreased LIF and IL-4 cytokines of decidual T cells are associated with unexplained recurrent abortions [47]. Moreover, the polymorphisms of *LIF* and *VEGF* genes are associated with litter size traits in pigs [10,14]. This evidence suggested that the *IL-4* gene may contribute to embryonic implantation and be related to litter size traits in pigs.

The porcine *IL-4R* gene is mapped on SSC3 at position 19.5 Mb and is closely located to the QTL regions for total number born (15.3–20.0 Mb) and some reproductive performance traits (11.9–33.7 Mb) in pigs [20,22]. It has recently been reported that the porcine *IL-4R* gene is involved in the gene-transcription factor networks for TNB trait in pigs [32]. Moreover, the *IL-4R* mRNAs are expressed in the endometrium, embryonic disc, and trophectoderm in pigs [30,31]. This evidence indicates that the *IL-4R* gene plays an important role in embryo implantation and may be related to litter size traits in pigs.

In this study, the porcine *IL-4R* c.1577A > T polymorphism was significantly associated with the NBA and NWA traits. A positive effect of the favorable *IL-4R* c.1577T allele on litter size traits was found. Remarkably, the porcine *IL-4R* c.1577A > T polymorphism revealed a non-synonymous mutation and exhibited a changing amino acid residue at position Q526L in the cytoplasmic domain of the IL-4R structural molecule. This domain region is important to the IL-4 signaling transduction response and exhibits several key functions to regulate the tyrosine phosphorylation and intracellular signal transduction pathways. It contains numerous tyrosine (Y) residues (at positions Y504, Y583, Y610, Y638, and Y718) and sequence motifs of the binding sites for the insulin receptor substrate-1 (IRS-1), IRS-2, and signal transducer and activator of transcription 6 (STAT6) in terms of the growth and gene expression of the regulation domains [50]. Moreover, a high degree of homology of the porcine IL-4R tyrosine residues and their surrounding amino acid residues with human IL-4R amino acid sequence [16,50] was observed in this study (Figure 3). It can be expected that the porcine IL-4R molecule has functions that are similar to the human IL-4R molecule. Therefore, the changing of Q526L was located in the growing domain of the cytoplasmic region that was close to the tyrosine-phosphorylation of the IRS sequence motif and may have affected the intracellular signal transduction pathways and the function of cells. Furthermore, four specific DNA motifs (^1419^CCCAGAGCC^1427^, ^1448^CCCAGAGCC^1456^, ^1448^CCCAGAGCT^1456^, and ^1569^CCCAGAGCT^1577^) were observed in the cytoplasmic region of the porcine *IL-4R* gene. Interestingly, the polymorphism of *IL-4R* c.1577A > T (Q526L) was also located at the nucleotide position 1577 on the ^1569^CCCAGAGCT^1577^ motif of the porcine *IL-4R* mRNA sequence (Genbank accession no. NM_214340.1). These sequence motifs might have affected its expression or relationships to functional variations. However, the variant Q526L of the porcine *IL-4R* gene has not yet been functionally identified. It has been hypothesized that the observed amino acid change of Q526L might serve as the functional explanation of these traits, although the possibility of mutations inside or outside the coding region in close linkage disequilibrium with porcine *IL-4R* c.1577A > T (Q526L) should not be excluded.

There is limited literature on the association of the *IL-4R* gene with litter size traits in pigs. However, several studies have demonstrated that the polymorphisms of *IL4R* gene are associated with the recurrent abortion, spontaneous preterm birth, and pre-eclampsia in humans [27,33,34,35]. Moreover, the decreased circulating IL-4R protein levels in serum were associated with pre-eclampsia in Danish women [51]. In contrast, no association of *IL-4R* variants with the recurrent abortion has been reported in Iranian women [52]. Furthermore, the polymorphisms of the *IL-4R* gene are associated with *LIF* mRNAs and protein expression levels [48] and variations of *LIF* gene are association with the recurrent abortions in women [47] and litter size traits in sows [10]. This evidence suggests that the *IL-4R* gene may be implicated in litter size traits in pigs.

In order to establish an elite breeding stock with high prolificacy traits in pigs, there is an essential procedure to accumulate identified genes or combine more affected genes from multiple parents into a superior genotype of the breeding stock, which is called gene pyramiding schema [53]. The gene pyramiding approach can be achieved through improving genetic values with marker-assisted selection (MAS) to accumulate favorable alleles from different loci for increasing the frequency of favorable alleles of genes controlling trait of interest in a population [53]. Thus, identification of the effects of combined genotypes or accumulated favorable alleles on litter size traits is required. There are numerous studies demonstrating that the accumulated favorable alleles, combined genotypes, or epistatic interactions of two or more loci QTLs or candidate genes have more beneficial effects on litter size traits in pigs [10,12,54,55].

Several studies have reported that the interactions of the *IL-4* and *IL-4R* genotypes are associated with diabetes, cancer, and pemphigus foliaceus in humans [43,56,57]. Unfortunately, no interaction effects of porcine *IL-4* and *IL-4R* genotypes on litter size traits were observed in this study (data not shown). Therefore, it was explored whether the accumulated favorable alleles of the porcine *IL-4* and *IL-4R* genotypes are associated with litter size traits in pigs. Interestingly, the significant effect of the accumulated favorable alleles (*IL-4* g.134993898T and *IL-4R* c.1577T) of porcine *IL-4* and *IL-4R* genes on litter size traits was exhibited. The increased numbers of favorable alleles were positively correlated with the TNB, NBA, and NWA traits in sows (Figure 2). Moreover, the accumulated favorable alleles (TTTT) were characterized as an advantageous genotype with the highest values for the NBA, NWA, and MWW traits. Meanwhile, the accumulated unfavorable alleles (CCAA) were characterized as a disadvantageous genotype with the lowest values for these traits (Table 6). This evidence indicates that there are strong additive effects of the accumulated favorable alleles of porcine *IL-4* and *IL-4R* genes on the litter size traits and the increasing numbers of favorable alleles seem to enhance litter size traits in these sows. Therefore, the accumulated favorable alleles (TTTT) can be used in marker-assisted selection to select the individuals with higher litter size traits. These findings suggest the gene pyramiding effects of porcine *IL-4* and *IL-4R* genes may provide advantages for the breeding industry. The results in this study indicate that the porcine *IL-4* and *IL-4R* genes could be expected to be involved in the reproductive processes of pigs, especially with regard to litter size traits. Further studies are required to confirm the association of these SNPs with litter size traits in larger population samples and various commercial pig breeds.

## 5. Conclusions

In this study, we found that polymorphisms of porcine *IL-4* and *IL-4R* genes are associated with the reproductive traits of pigs. The porcine *IL-4* g.134993898T > C and *IL-4R* c.1577A > T polymorphisms had clear effects on NBA, NWA, and MWW traits in commercial pigs. Moreover, the accumulation of favorable alleles of porcine *IL-4* and *IL-4R* genes is positively correlated with the litter size traits. These findings emphasize the importance of the porcine *IL-4* and *IL-4R* genes in the reproductive traits of pigs. Therefore, the porcine *IL-4* and *IL-4R* genes may be potential candidate genes for the genetic improvement of litter size traits in the pig breeding industry.

## Figures and Tables

**Figure 1 animals-11-01154-f001:**
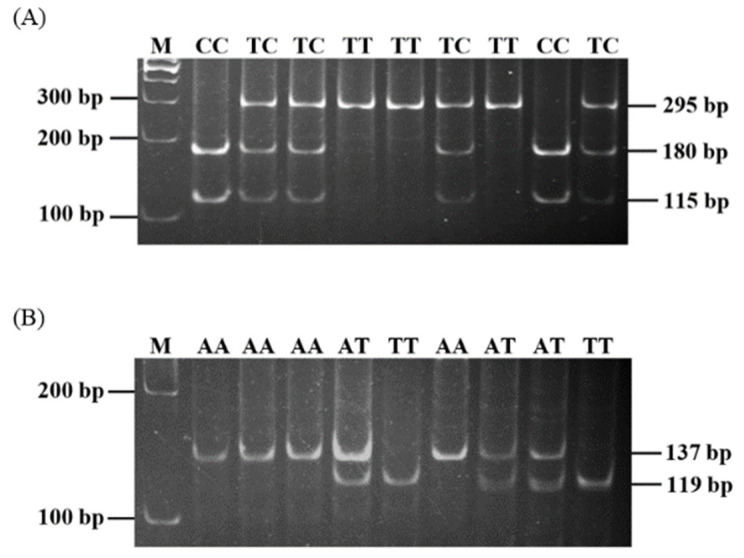
Genotyping single nucleotide polymorphisms (SNPs) of porcine *IL-4* and *IL-4R* genes (**A**) at *IL-4* g.134993898T > C locus with *Bsu*RI and (**B**) at *IL-4R* c.1577A > T locus with *Alu*I. The molecular marker of 100 bp DNA ladder (M) and the genotypes of porcine *IL-4* (CC, TC, and TT) and *IL-4R* (AA, AT, and TT) genes are indicated at the top of each line. IL, interleukin.

**Figure 2 animals-11-01154-f002:**
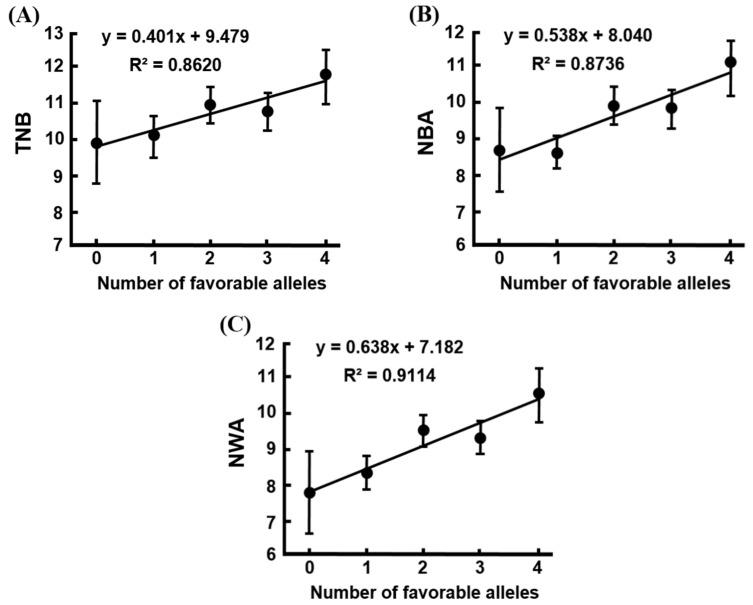
Linear regression analysis of the number of favorable alleles of porcine *IL-4* and *IL-4R* genes and least square mean values of (**A**) TNB: total number born, (**B**) NBA: number born alive, and (**C**) NWA: number of piglets weaned alive traits in later parities of pigs. The y indicates means phenotypic values of litter size traits, x indicates the number of accumulated favorable alleles, and R^2^ indicates the coefficient of determination for the linear regression equation.

**Figure 3 animals-11-01154-f003:**
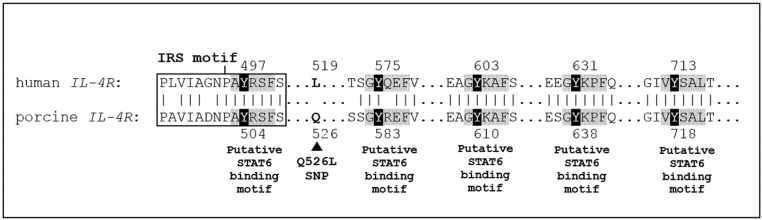
Alignment of porcine IL-4R amino acid sequence (GenBank Accession no. NP_999505.1) and human IL-4R amino acid sequence (NP_000409.1). Tyrosine (Y) residues of porcine IL-4R amino acid sequence are localized in the cytoplasmic region at positions Y504, Y583, Y610, Y638, and Y718, corresponding to human IL-4R sequence at positions Y497, Y575, Y603, Y631, and Y713. Sequence motifs of binding sites for insulin receptor substrate (IRS) and signal transducer and activator of transcription 6 (STAT6) are indicated by boxed and shaded areas. | = identity, ▲ = the porcine IL-4R polymorphism at position Q526L (c.1577A > T). SNP, single nucleotide polymorphism.

**Table 1 animals-11-01154-t001:** Primer sequences used for single nucleotide polymorphism (SNP) identification in porcine *IL-4* gene. PCR, polymerase chain reaction; IL, interleukin; Ta, annealing temperature.

Primers	Location	Primer Sequences	PCR Size (bp)	Ta (°C)
*IL-4*-1	Exon 1	F: 5′-GTTAGCTCCTCCCAGTAAAC-3′ R: 5′-CATCAGAGCATCTGGAGAGA-3′	236	60
*IL-4*-2	Exon 2	F: 5′-AACGCTCTCTCTTGCTCTCT-3′ R: 5′-TGCCTCCTGTATTCCAGGTA-3′	409	60
*IL-4*-3	Exon 3	F: 5′-ACACGGTAATTGGTGGCTCT-3′ R: 5′-TGTTAGAAGCCAAGCTGTGG-3′	189	60
*IL-4*-4	Exon 4	F: 5′-AGGGAGAAAGGCATGAGCAA-3′ R: 5′-GCCTATGCCATTAAAGTGACA-3′	227	60
*IL-4*-5	Intron 2	F: 5′-GTACACGCCCTCTTCTAAGA-3′ R: 5′-TCCCTGGAAATCTCACATGC-3′	308	60
*IL-4*-6	Intron 3	F: 5′-GTGGTAGGTATCCTTTCCAC-3′ R: 5′-AACAGTGATCAAACCAGGGC-3′	333	58

**Table 2 animals-11-01154-t002:** Primer sequences and restriction enzymes used for genotyping of porcine *IL-4* and *IL-4R* genes. PCR, polymerase chain reaction; IL, interleukin; SNP, single nucleotide polymorphism; Ta, annealing temperature.

SNP Position (SNP ID)	Location	Primer Sequences	PCR Size (bp)	Ta (°C)	Restriction Enzyme
*IL-4* g.134993898T > C (rs329453960)	Intron 3	F: 5′-GTGGTAGGTATCCTTTCCAC-3′ R: 5′-AACAGTGATCAAACCAGGGC-3′	333	58	*Bsu*RI
*IL-4R* c.163G > A (rs342744061)	Exon 1	F: 5′-TCTTGATCACTGGGCTTCCG-3′ R: 5′-AACTCAGCGCTGCAGTTGAC-3′	152	60	*Pfl*FI
*IL-4R* c.242C > T (rs334778260)	Exon 2	F: 5′-GGTCACATGACCAGCCTAAT-3′ R: 5′-TTGAAGGAGCTGTTCCACAG-3′	180	60	*Afl*III
*IL-4R* c.623G > A (rs790596006)	Exon 4	F: 5′-TCTATAACGTGACCTACCTG-3′ R: 5′-TTAAGCCACTTGACACTCGG-3′	148	60	*Mlu*I
*IL-4R* c.1016G > T (rs692527061)	Exon 8	F: 5′-GCTGGAAGACTTGTCTTACC-3′ R: 5′-GATCGTCTTGCTGACCTCTA-3′	149	60	*Bsu*RI
*IL-4R* c.1577A > T (rs342791614)	Exon 8	F: 5′-CTGGACTCGGACCCAGAG-3′ R: 5′-ACACTCTGGCGCAGGATCT-3′	137	58	*Alu*I

**Table 3 animals-11-01154-t003:** Genotypic and allelic frequencies of porcine *IL-4* and *IL-4R* genes in Landrace sows.

SNPs	*n*	Genotypic Frequencies			Allelic Frequencies ^1^		*p*-Value ^2^ (χ^2^)
		AA	AB	BB	A	B	
*IL-4* g.134993898T > C	320	0.41	0.47	0.12	0.65	0.35	0.90
*IL-4R* c.1577A > T	318	0.28	0.55	0.17	0.56	0.44	0.10

^1^ Allele A represents wild type alleles of the *IL-4* g.134993898T and *IL-4R* c.1577A for each locus, and allele B represents mutate alleles of *IL-4* g.134993898C and *IL-4R* c.1577T. ^2^ The *p*-value is considered a significant level of the chi-square (χ^2^) test for Hardy–Weinberg equilibrium of each locus. SNPs, single nucleotide polymorphisms.

**Table 4 animals-11-01154-t004:** Association of porcine *IL-4* g.134993898T > C locus with litter size traits.

Parity	Traits ^1^	Genotypes (Means ± SE) ^2^			Additive	Dominance
		TT	TC	CC		
First parity	*n*	132	150	38		
	TNB	9.58 ± 0.45	9.77 ± 0.48	9.68 ± 0.67	−0.05 ± 0.25	0.14 ± 0.32
	NBA	8.65 ± 0.47	8.57 ± 0.36	8.08 ± 0.42	0.29 ± 0.27	0.11 ± 0.12
	NWA	8.15 ± 0.47	7.85 ± 0.64	7.28 ± 0.54	0.44 ± 0.22	0.14 ± 0.12
	MBW	1.58 ± 0.06	1.60 ± 0.08	1.67 ± 0.08	−0.04 ± 0.02	−0.02 ± 0.02
	MWW	6.42 ± 0.12	6.58 ± 0.15	6.52 ± 0.18	−0.05 ± 0.05	0.11 ± 0.08
Later parities (2nd–8th parities)	*n*	348	380	97		
	TNB	11.25 ± 0.51	11.03 ± 0.53	10.72 ± 0.55	0.27 ± 0.28	0.04 ± 0.17
	NBA	10.48 ± 0.58	9.92 ± 0.57	9.47 ± 0.75	0.51 ± 0.32	0.06 ± 0.32
	NWA	9.72 ± 0.47 ^b^	9.42 ± 0.47 ^b^	8.45 ± 0.52 ^a^	0.64 ± 0.21 *	0.33 ± 0.34
	MBW	1.58 ± 0.05	1.57 ± 0.04	1.54 ± 0.05	0.02 ± 0.01	0.01 ± 0.02
	MWW	6.62 ± 0.05	6.67 ± 0.05	6.59 ± 0.07	0.02 ± 0.02	0.06 ± 0.02

^1^*n*: number of investigated litters, TNB: total number born, NBA: number born alive, NWA: number of piglets weaned alive, MBW: mean birth weight of the piglets, MWW: mean weight of piglets at weaning. MBW and MWW traits are presented in kg. ^2^ Means ± SE represents least square means ± standard error. TT, TC, and CC are genotypes of porcine *IL-4* g.134993898T > C locus. Values in each row with differing superscripts are considered significantly different (^a,b^
*p* < 0.05), * *p* < 0.05.

**Table 5 animals-11-01154-t005:** Association of porcine *IL-4R* c.1577A > T locus with litter size traits.

Parity	Traits ^1^	Genotypes (Means ± SE) ^2^			Additive	Dominance
		AA	AT	TT		
First parity	*n*	89	176	53		
	TNB	9.48 ± 0.38	10.03 ± 0.76	9.27 ± 0.58	0.11 ± 0.32	0.65 ± 0.35
	NBA	8.25 ± 0.37	8.55 ± 0.52	8.19 ± 0.67	0.03 ± 0.32	0.33 ± 0.47
	NWA	7.45 ± 0.53	7.57 ± 0.61	7.30 ± 0.49	0.08 ± 0.35	0.20 ± 0.42
	MBW	1.54 ± 0.04	1.58 ± 0.08	1.67 ± 0.08	−0.06 ± 0.02	−0.02 ± 0.03
	MWW	6.42 ± 0.12	6.39 ± 0.10	6.55 ± 0.18	−0.06 ± 0.05	−0.09 ± 0.07
Later parities (2nd–8th parities)	*n*	235	449	133		
	TNB	10.51 ± 0.49	10.55 ± 0.43	11.54 ± 0.68	−0.52 ± 0.28	−0.47 ± 0.32
	NBA	9.21 ± 0.56 ^a^	9.43 ± 0.47 ^a^	10.71 ± 0.64 ^b^	−0.75 ± 0.28 *	−0.53 ± 0.36
	NWA	8.61 ± 0.52 ^a^	9.09 ± 0.46 ^a^	10.20 ± 0.58 ^b^	−0.80 ± 0.28 **	−0.32 ± 0.31
	MBW	1.55 ± 0.05	1.60 ± 0.04	1.57 ± 0.05	−0.01 ± 0.02	0.04 ± 0.02
	MWW	6.62 ± 0.06	6.63 ± 0.05	6.60 ± 0.06	0.01 ± 0.02	0.02 ± 0.03

^1^*n*: number of investigated litters, TNB: total number born, NBA: number born alive, NWA: number of piglets weaned alive, MBW: mean birth weight of the piglets, MWW: mean weight of piglets at weaning. MBW and MWW traits are presented in kg. ^2^ Means ± SE represents least square means ± standard error. AA, AT, and TT are genotypes of porcine *IL-4R* c.1577A > T locus. Values in each row with differing superscripts are considered significantly different (^a,b^
*p* < 0.05), * *p* < 0.05, ** *p* < 0.01.

**Table 6 animals-11-01154-t006:** Association of accumulated favorable alleles of porcine *IL-4* g.134993898T > C and *IL-4R* c.1577A > T loci, with litter size traits.

Parity	Traits ^1^	Number of Favorable Alleles (Means ± SE) ^2^				
		0	1	2	3	4
First parity	*n*	7	59	144	89	18
	TNB	10.72 ± 1.42	9.58 ± 0.57	10.12 ± 0.54	10.15 ± 0.52	8.68 ± 0.72
	NBA	8.84 ± 1.25	8.24 ± 0.58	8.74 ± 0.52	9.15 ± 0.67	8.12 ± 0.95
	NWA	7.23 ± 1.27	7.25 ± 0.73	7.88 ± 0.64	8.57 ± 0.71	7.82 ± 0.84
	MBW	1.48 ± 0.16	1.58 ± 0.05	1.57 ± 0.04	1.56 ± 0.05	1.58 ± 0.08
	MWW	6.52 ± 0.32	6.41 ± 0.12	6.55 ± 0.10	6.43 ± 0.08	6.64 ± 0.19
Later parities (2nd–8th parities)	*n*	18	152	371	225	48
	TNB	9.94 ± 1.14	10.21 ± 0.62	10.87 ± 0.51	10.68 ± 0.43	11.71 ± 0.78
	NBA	8.81 ± 1.17 ^ab^	8.75 ± 0.55 ^a^	9.91 ± 0.46 ^b^	9.85 ± 0.47 ^b^	10.95 ± 0.77 ^b^
	NWA	7.81 ± 1.11 ^ab^	8.34 ± 0.55 ^a^	9.51 ± 0.43 ^bc^	9.30 ± 0.47 ^bc^	10.52 ± 0.72 ^c^
	MBW	1.65 ± 0.07	1.61 ± 0.05	1.63 ± 0.04	1.66 ± 0.05	1.58 ± 0.07
	MWW	6.21 ± 0.11 ^a^	6.64 ± 0.04 ^b^	6.65 ± 0.03 ^b^	6.63 ± 0.03 ^b^	6.65 ± 0.05 ^b^

^1^*n*: number of investigated litters, TNB: total number born, NBA: number born alive, NWA: number of piglets weaned alive, MBW: mean birth weight of the piglets, MWW: mean weight of piglets at weaning. MBW and MWW traits are presented in kg. ^2^ Means ± SE represents least square means ± standard error. Values in each row with differing superscripts are considered significantly different (^a,b,c^
*p* < 0.05). Number of favorable alleles is accumulated favorable alleles of the porcine *IL-4* g.134993898T and *IL-4R* c.1577T alleles.

## Data Availability

The data presented in this study are not publicly available due to privacy restrictions.

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
