# Peer review of "Association of IL-4 and IL-4R Polymorphisms with Litter Size Traits in Pigs"

_animals, 2021, doi:10.3390/ani11041154_

Round 1

Reviewer 1 Report

The conducted research concerns the possibility of improving the breeding and performance value of pigs in relation to the breeding performance characteristics. They are one of many attempts to determine genetic markers for the number of piglets and their body weight. In general, the analyzes were performed correctly, in accordance with generally accepted rules.

Comments:

  1. Material and methods

„All sows were reared under commercial conditions of the Betagro Hybrid International Company”

- please provide more details for better readers' understanding of the text. Environmental conditions and system of maintenance and nutrition are the main factors determining the level of reproductive traits.

„The reproductive performance traits of the sows were assessed in 1162 litters and were recorded in terms of litter size”

- please specify which reproductive cycles were analyzed: as shown in Table 4 from the first to eighth. Please provide this information in the Material and methods

  1. Conclusions

„Therefore, these two SNPs could be used as candidate genes for the genetic improvement of litter size traits in the pig breeding industry”

- I believe that based on the results obtained, no such conclusion should be drawn. The analysis concerns 323 Landrace sows, which is too little experimental material for one of the most world's numerous pig breeds in the world. This does not diminish the obtained results, but should be verified based on a larger number of sows and litters. The research results provide an excellent basis, but should be considered preliminary.

References

- please reduce the number of references cited, information is often repeated.

Author Response

Reviewer Response

Reviewer 1:

Comments and Suggestions for Authors 

The conducted research concerns the possibility of improving the breeding and performance value of pigs in relation to the breeding performance characteristics. They are one of many attempts to determine genetic markers for the number of piglets and their body weight. In general, the analyzes were performed correctly, in accordance with generally accepted rules.

Comments:

  1. Material and methods

2.1 „All sows were reared under commercial conditions of the Betagro Hybrid International Company” please provide more details for better readers' understanding of the text. Environmental conditions and system of maintenance and nutrition are the main factors determining the level of reproductive traits.

Response: All sows were reared under commercial conditions of the Betagro Hybrid International Company, Thailand. These sows were kept in the closed houses with an evaporative cooling system and were fed a corn-soybean-based diet containing 16% crude protein and 3388 kcal/kg digestible energy.

2.2 „The reproductive performance traits of the sows were assessed in 1162 litters and were recorded in terms of litter size”

- please specify which reproductive cycles were analyzed: as shown in Table 4 from the first to eighth. Please provide this information in the Material and methods

Response: The parity information has been added as “The reproductive performance traits of the sows were assessed in 1162 litters (1 to 8 parities) and were recorded in terms of litter size.”.

  1. Conclusions

„Therefore, these two SNPs could be used as candidate genes for the genetic improvement of litter size traits in the pig breeding industry”

- I believe that based on the results obtained, no such conclusion should be drawn. The analysis concerns 323 Landrace sows, which is too little experimental material for one of the most world's numerous pig breeds in the world. This does not diminish the obtained results, but should be verified based on a larger number of sows and litters. The research results provide an excellent basis, but should be considered preliminary.

Response: Thank you for your suggestion, the conclusion has been revised as Lines 429-431: “Therefore, the porcine IL-4 and IL-4R genes may be potential candidate genes for the genetic improvement of litter size traits in the pig breeding industry.”. 

In addition, verified information has been added as Lines 420-422: “Further studies are required to confirm the association of these SNPs with litter size traits in larger population samples and various commercial pig breeds.”.

References – please reduce the number of references cited, information is often repeated.

Response: The number of references has been reduced. Moreover,the repeated information has been revised and reduced.

Thank you for your valuable suggestion.

Reviewer 2 Report

This manuscript describes a study identifying associations between IL-4 and IL-4 receptor alleles and reproductive traits in swine.  The paper is well written, and the study is mostly well described.  Some suggestions for improvement are offered below.

Authors describe how these loci are closely linked with other QTL for reproductive traits, but then make a big deal about the observed associations.  It is not evident that the causative mutation has been identified, and comparing the use of IL-4 or IL-4R as markers for MAS to the previously identified QTL would strengthen the argument for the value of this work. 

The authors use two papers to indicate the economic relevance of litter size traits.  It is curious that the two cited works are using Berkshire pigs, known for small litters, while the subjects of the present research are Landrace, among the highest litter size breeds. 

What is the selection history of the test population?  Can some information on the selection index or traits selected for be presented?  Presumably litter size born or weaned has been under selection at some level.

The statistical analysis section should be developed.  Specifically, how did authors handle the fact that most of the traits are not independent?  e.g. TNB, NBA, NWA, MBW and MWW are all correlated traits, so assuming independence in the analysis will not appropriately identify significance of differences.

Table 4: one of the 20 comparisons was significant at the P<0.05.  That is the rate of false positives. 

Author Response

Reviewer Response

Reviewer 2:

Comments and Suggestions for Authors

This manuscript describes a study identifying associations between IL-4 and IL-4 receptor alleles and reproductive traits in swine. The paper is well written, and the study is mostly well described. Some suggestions for improvement are offered below.

  1. Authors describe how these loci are closely linked with other QTL for reproductive traits, but then make a big deal about the observed associations. It is not evident that the causative mutation has been identified, and comparing the use of IL-4 or IL4R as markers for MAS to the previously identified QTL would strengthen the argument for the value of this work.

Response: The results in this study provided evidence to indicate that genetic variants in the porcine IL-4 and IL-4R genes exist that affect the litter size traits in Landrace pigs. These findings promote the importance of the porcine IL-4 and IL-4R genes as candidate genes for the genetic improvement of litter size traits in pig breeding.   

  1. The authors use two papers to indicate the economic relevance of litter size traits. It is curious that the two cited works are using Berkshire pigs, known for small litters, while the subjects of the present research are Landrace, among the highest litter size breeds.

Response: New pieces of literature of the economic relevance of litter size traits in several pig breeds (Large White, Yorkshire, Landrace) have been added as below:

  • Metodev, S.; Thekkoot, D.M.; Young, J.M.; Onteru, S.; Rothschild, M.F.; Dekkers, J.C.M. A whole-genome association study for litter size and litter weight traits in pigs. Livest. Sci. 2018, 211, 87-97.
  • Schneider, J.F.; Rempel, L.A.; Snelling, W.M.; Weidmann, R.T.; Nonneman, D.J.; Rohrer, G.A. Genome-wide association study of swine farrowing traits. Part II: Bayesian analysis of marker data. J. Anim. Sci. 2012, 90, 3360-3367.

  1. What is the selection history of the test population? Can some information on the selection index or traits selected to be presented? Presumably, litter size born or weaned has been under selection at some level.

Response:  These Landrace sows are breeding stocks that were obtained from a commercial nucleus herd that is established for improved growth performance and reproductive traits by conventional method during two decades. The selection index cannot be published. The means and standard deviation of phenotype data for TNB, NBA, and NWA traits were 11.17+2.77, 9.58+2.98, and 8.99+2.25, respectively as well as for MBW and MWW traits were 1.58+0.20 and 6.70+0.32 kg, respectively (unpublished data).  

  1. The statistical analysis section should be developed. Specifically, how did authors handle the fact that most of the traits are not independent? e.g. TNB, NBA, NWA, MBW, and MWW are all correlated traits, so assuming independence in the analysis will not appropriately identify the significance of concern differences.

Response: Thank you very much for your valuable comments. We agree with your concern about these issues. Indeed, our data showed a highly positive correlation between TNB, NBA, and NWA traits (r = 0.68 to 0.92, p<0.001). Moreover, the MBW was positive correlated with MMW trait (r=0.12, p = 0.002). In contrast, the TNB, NBA, and NWA traits were negative correlated with MBW and MWW traits (r = -0.15 to -0.40, p<0.01). Therefore, we try to improve the statistical model with included TNB, NBA, or NWA as covariate factors in several models for analyzing the effects of IL4 and IL-4R genes on the NBA (included TNA as a covariate), NWA (included TNB and/or NBA as covariate factors), MBW (included TNB and/or NBA as covariate factors) and MWW (included TNB and/or NBA and/or NWA as covariate factors). Although, these covariate factors revealed a significant effect on the litter size traits. However, their effects cannot enhance the power of markers in the model analysis to detect significance on the litter size traits in this investigation. Therefore, we have chosen the statistical model as described in the statistical analysis section to explore whether the porcine IL-4 and IL-4R polymorphisms are associated with litter size traits in pigs. 

  1. Table 4: one of the 20 comparisons was significant at the P<0.05. That is the rate of false positives.

Response: It is possible. Indeed, the result of this study demonstrated that the porcine IL-4 polymorphism was associated with the NWA trait as well as showed a trend toward association with NBA trait (p = 0.06) in this Landrace pig population.

Thank you for your valuable suggestion.
